# Flexible Conductive Polymer Film Grafted with Azo-Moieties and Patterned by Light Illumination with Anisotropic Conductivity

**DOI:** 10.3390/polym11111856

**Published:** 2019-11-11

**Authors:** Yevgeniya Kalachyova, Olga Guselnikova, Vladimir Hnatowicz, Pavel Postnikov, Vaclav Švorčík, Oleksiy Lyutakov

**Affiliations:** 1Department of Solid State Engineering, University of Chemistry and Technology, 16628 Prague, Czech Republic; yevgeniya.kalachyova@vscht.cz (Y.K.); postnikov@tpu.ru (P.P.); vaclav.svorcik@vscht.cz (V.Š.); 2Research School of Chemistry and Applied Biomedical Sciences, Tomsk Polytechnic University, 634049 Tomsk, Russian Federation; 3Nuclear Physics Institute, Academy of Sciences of the Czech Republic, 25068 Rez, Czech Republic; hnatowicz@ujf.cas.cz

**Keywords:** PEDOT:PSS, flexible film, light pattering, electrical properties, resistance anisotropy

## Abstract

In this work, we present the method for the creation of an anisotropic electric pattern on thin poly(3,4-ethylenedioxythiophene): polystyrene sulfonate (PEDOT:PSS) films through PSS grafting by azo-containing moieties followed by light-induced polymers redistribution. Thin PEDOT:PSS films were deposited on the flexible and biodegradable polylactic acid (PLLA) substrates. The light-sensitive azo-groups were grafted to PSS using the diazonium chemistry followed by annealing in methanol. Local illumination of azo-grafted PEDOT:PSS films through the lithographic mask led to the conversion of azo-moieties in Z-configuration and further creation of the lateral gradient of azo-isomers along the film surface. The concentration gradient led to the migration of PSS away from the illuminated area, increasing the PEDOT chains’ concentration and the corresponding increase of local electrical conductivity in the illuminated place. Utilization of mask with linear pattern results in the appearance of conductive PEDOT-rich and non-conductive PSS-rich lines on the film surface, and final, lateral anisotropy of electric properties. Our work gives an optical lithography-based alternative to common methods for the creation of anisotropic electric properties, based on the spatial confinement of conductive polymer structures or their mechanical strains.

## 1. Introduction

Conductive polymer (CP) composites are receiving a great deal of attention owing to their unique electrical, mechanical, and processing properties [1,2]. The combination of flexibility with conductive properties of the thin CPs films makes these materials unique and unparalleled in the world [3,4]. The as-prepared CPs or their composites exhibit isotropic electrical conductivity, which is due to the random dispersion of conductive pathways in polymer matrix [5]. Recently, CPs with conductive anisotropy have deserved special attention since they appear to be more suitable for applications in a range of electronic and optoelectronic devices [6,7,8,9,10]. The introduction of anisotropic conductivity into CPs seems to offer immense potential for diverse technological possibilities in a broad spectrum of devices, such as organic solar cells, flexible sensing devices, antistatic coatings, batteries, and electrodes for future energy sources [11,12,13,14].

The anisotropic conductivity can arise from the geometrical arrangement of conductive pathways in CPs composites or from the mobility anisotropy of charge carriers in the intrinsically conjugated polymers [15,16,17]. In the latter case, the anisotropy appears as a result of the ordered alignment of polymers chains or crystal domains [18,19]. Such an orientation of flexible polymer chains (or their composites) with anisotropic conductivity can be realized through the specific conditions of preparation or by post-preparation methods, such as the treatment by mechanical stretching or by direct or alternating electromagnetic field [20,21,22,23,24,25]. However, in most cases, the degree of anisotropic conductivity does not exceed one order of magnitude [26,27]. An alternative method for the introduction of anisotropic conductivity to CPs lies in the patterning of their thin films [28]. Using this approach, significant resistance anisotropy, larger than four orders of magnitude, was obtained [29]. However, such an approach is complicated by the natural properties of CPs, which are not suitable candidates for processing by soft- or photo-lithography [30,31,32,33].

Among the CPs family, the poly(3,4-ethylenedioxythiophene) (PEDOT) exhibits excellent electrical, optical, and mechanical properties [34]. The creation of anisotropic conductivity in PEDOT films was demonstrated recently through the PEDOT doping with anisotropic particles [35,36]. Recently, we demonstrated the ability of the local tuning of PEDOT:PSS local conductivity in a great range through the incorporation of light-isomerized chemical moieties and light-induced migration of PSS chains from the illuminated area and local increase of PEDOT concentration [37,38]. In this work, we extended the above-mentioned approach on the preparation of the PEDOT:PSS film with anisotropy in a surface electrical conductivity by a simple and scalable procedure. For introduction of anisotropic conductivity we used the light, which makes this the first report of the application of such stimuli in the case of CPs (light was previously used for creation of anisotropic properties in the case of liquid crystal [39,40] but was never reported for realization of such a goal in the CP field).

## 2. Experimental

### 2.1. Materials

A poly-L-lactic acid thin films (PLLA, thickness 25 µm) were supplied from Goodfellow, Ltd. (Huntingdon, UK). Poly(3,4-ethylene dioxythiophene):poly(styrenesulfonate) (PEDOT:PSS) water suspension was supplied from the Sigma-Aldrich (Prague, Czech Republic); 4-nitrobenzenediazonium tosylate (ADT-NO_2_) was synthesized according to the published procedure [41].

### 2.2. Sample Preparation

The PLLA thin films (1 × 2 cm^2^) were cleaned by washing in deionized water and methanol and subsequently dried. PEDOT:PSS aqueous solution was deposited by spin coating at 500 rpm for 10 min on PLLA films’ surfaces and thermally annealed for 5 h at 40 °C under an argon atmosphere (the thickness of PEDOT:PSS films, estimated from control measurements, using the scratch tests and microscopic glass as substrate—approximately 800 nm). Dried films were modified by immersion of samples in 1 mM aqueous solution of ADT-NO_2_ for 20 min, subsequent aging at 50 °C for 12 h, and washing by deionized water. The sample annealing was performed through immersion in methanol for 5 min, followed by holding the samples in the saturated vapor of methanol at 40 °C for 20 min. Sample illumination was performed via a lithographic mask (40 µm wide transparent lines separated by 40 µm wide opaque areas) with the utilization of LED light source (375 nm, 1275 mW, and light spot—3 cm^2^) for 60 min.

### 2.3. Sample Characterization

UV–Vis absorption spectra were measured using Lambda 25 UV/Vis/NIR Spectrometer (Waltham, MA, USA). ATR-IR were obtained using the Nicolet iS 10 FT-IR Spectrometer (Nicolet, Prague, Czech Republic). X-ray photoelectron spectroscopy (XPS) was measured using an Omicron Nanotechnology ESCAProbeP spectrometer (Omicron Nanotechnology Ltd., Taunusstein, Germany) fitted with monochromatic Al K Alpha X-ray source at 1486.7 eV. Surface profiles were taken using the Hommel 1000 profilometer (standard deviation ± 10%). The surface morphology and conductivity were measured using the Icon (Bruker, Karlsruhe, Germany) atomic force microscope in Tuna current mode with Pt/Pd coated silicon tip. For the measurements of the sheet resistance and current-voltage (CV) characteristics using a picoampermeter, KEITHLEY 487 (Cleveland, OH, USA), two planar gold electrodes (thickness 50 nm) were evaporated on sample edges. The current–voltage characteristics of samples were measured in a −0.25–0.25 V voltage range using the manual tuning of applied potential. The temperature dependencies of samples resistance were measured in a −25–25 °C temperature interval using the climatic chamber.

## 3. Results and Discussion

The schematic representation of the present experiment, comprised of the introduction of light-sensitivity to the PEDOT:PSS and the light-induced formation of anisotropic conductive pattern, is presented in Figure 1. First, thin PEDOT:PSS films were deposited on the flexible PLLA substrates (25 µm thick). Then diazonium modification was carried out, according to the previously described procedure [37], in order to graft the 4-nitroazophenylen moieties to PSS chains (grafted samples were further referred to as PEDOT:PSS-NAP). The 4-nitroazophenylene moieties can undergo the light-induced isomerization under the illumination with a wavelength corresponding to *E-* or *Z*-isomer. Our previous experiments [37,38] have shown that the grafting of PSS with 4-nitroazobenzene leads to the appearance of both steric isomers, so that here, the vapor annealing in the methanol vapors was applied to induce the formation of the more thermodynamically favorable form of *trans*-4-nitroazophenylene groups. In the next step, the samples were illuminated through the photolithographic mask (with periodically transparent and opaque lines) with a 375 nm wavelength, corresponding to the absorption band of *E*-4-nitroazobenzene isomer. Local light triggering results in the transition of 4-nitroazobenzene from *E*- to Z-isomers and the appearance of the concentration gradient between illuminated and mask-screened area. As a result of the concentration gradient, the driving force for the flow of PSS from illuminated to non-illuminated areas occurs and leads to the formation of periodical structure, containing the conductive PEDOT-rich (illuminated) and PEDOT-diluted (non-illuminated) areas.

The chemical modification of PSS chains and grafting of 4-nitroazobenzene moieties was checked by XPS and IR spectroscopy (Figure 2A–C). The XPS results (Figure 2A) indicate the appearance of nitrogen-related peaks (at 400.7 eV –N=N– and 405.9 eV –NO_2_) after 4-nitroazophenylene grafting with apparent split shape, where the positions of maxima correspond to nitrogen in –NO_2_ and –N=N– groups. The results of IR spectroscopy are presented in Figure 2B, and detailed peak affiliations are given in the Table 1. The grafting of 4-nitroazophenylene groups to PSS leads to the appearance of several IR absorption bands, corresponding well with 4-nitroazobenzene structure (Table 1). The relative intensity of the 4-nitroazophenylene IR absorption band is changed under light triggering (at 375 nm wavelength); such a change is typical for *E*/Z azo group isomerization (Figure 2C). Finally, Figure 2D shows the UV–Vis absorption spectra of PEDOT:PSS-NAP films before and after the light triggering at the same wavelength. In the pristine spectrum, the absorption peak visible near the 380 nm corresponds to that characteristic for *E*-azobenzene isomer π→π* electron transition. Light triggering induces the isomerization of azobenzene expressed by the disappearance of π→π* absorption band [42] and the appearance of a new band, corresponding to characteristic one for a *Z*-form of nitroazobenzene moieties (n→π* electron transition). So, the results of XPS, IR, and UV–Vis confirm the successful grafting of azophenylene moieties and the ability to induce their isomerization by illumination at a wavelength of 375 nm.

In the next step, the PEDOT:PSS-NAP samples were illuminated through the lithographic mask (40 µm wide transparent lines separated by 40 µm wide opaque ones). A microscopic image of the illuminated samples is presented in Figure 3A. The illumination leads to the appearance of periodical structure with periodicity corresponding well with the optical mask pattern. The white areas in the Figure 3A represent the light-triggered places on the PEDOT:PSS-NAP surface, while the dark areas correspond to the mask-screened, non-illuminated ones. The surface profiles of the samples, measured by the profilometry, are shown in Figure 3B. The appearance of a clearly visible periodical surface pattern proves real polymer-flow initiated by the local sample illumination. The polymer migration out of the illuminated places leads to the creation of valleys on the surface profile. Local light-triggering results in the transition of 4-nitroazobenzene from *E*- to Z-isomers and the appearance of the concentration gradient between the illuminated and mask-screened area. As a result of the concentration gradient, the driving force for the redistribution of polymer chains appears. Since the 4-nitroazobenzene is grafted solely to PSS chains, only this polymer is included in the migration process. In particular, there is a tendency to compensate the light-created isomers’ gradient with the PSS chains’ flow from the illuminated area to mask-screened (the similar phenomenon of light-induced azo-moieties isomerization and related material flow is widely described in the literature [43,44,45]). As a result, the PSS-rich and PEDOT-pure (mask screened) and PSS-pure and PEDOT-rich (illuminated) areas occur through the previously flat and homogeneous PEDOT:PSS thin film.

The periodicity of created pattern corresponds to the photolithographic mask used, while the amplitude (height difference) is approximately 700 nm. The height difference indicates the decrease of initial PEDOT:PSS thickness (800 nm, approximately determined from control measurements, using the microscopic glass as substrate and scratch-tests) by approximately 50% in the illuminated areas and an increase of polymer thickness in the screened areas by 50% as well.

The evolution of PEDOT:PSS-NAP thin film electrical resistance during the illumination through the lithographic mask is shown in Figure 3C, where the resistance is plotted as a function of illumination time. The experiments were performed in two arrangements, with electrodes located either along or perpendicularly to light-induced pattern on PEDOT:PSS-NAP films (see Figure 3C). As could be expected, the redistribution of polymers significantly affect the electrical properties of then PEDOT:PSS-NAP film, and an apparent resistance anisotropy is seen which increases with illumination time. In the case of the parallel electrode arrangement, the PEDOT-rich areas are separated by the PEDOT-depleted areas, and thin film resistance in this direction gradually increases with the illumination time. Oppositely, in the perpendicular electrodes’ arrangement, the planar gold electrodes are “connected” by PEDOT reach lines and the samples resistance in this case gradually decreases. All changes in the samples resistance were observed during the first 210 min of sample illumination; longer illumination does not lead to measurable resistance changes. The maximum value of resistivity anisotropy, more than two orders of magnitude, was found in this experiment. Assuming of the conservation of general material volumes (i.e., general PEDOT:PSS cross-section area), the resistivity values can be calculated as 3 × 10^−3^ for pristine, 5 × 10^−2^ for illuminated perpendicularly, and 2 × 10^−4^ for parallel illuminated samples.

The current–voltage characteristics of the pristine, anisotropic, PEDOT:PSS-NAP thin films are presented in Figure 3E–G (measurements were performed in −0.25–0.25 voltage range). The experiments were also performed in two electrodes arrangements, with simultaneous mechanical bending of the film (Figure 3D). As is evident from Figure 3E, the pristine PEDOT:PSS-NAP film exhibits ohmic behavior; i.e., linear current–voltage dependence. The same linear dependence is also observed in the case of PEDOT:PSS-NAP patterned perpendicularly to the electrodes (see Figure 3F). In the case of parallelly patterned PEDOT:PSS-NAP, however, an apparent deviation from linear current–voltage dependence is found, indicating an increasing impact of dielectric PSS-rich barriers, isolating the conductive PEDOT-rich pathways (Figure 3G). Since in the last experimental arrangement, the PEDOT-rich areas were separated by PEDOT-pure areas, the loop observed on the I–V curves (Figure 3G) can be attributed to increased conductivity of “barriers” layer due to charge carrier injections or formation of the additional conductive pathway under the application of higher voltage. Finally, it should be noted that the film bending does not affect the films’ electrical properties. The current–voltage dependencies for both electrode arrangements almost completely copy each other regardless of the bending degree. Such behavior may allow for the future application of the proposed material in various fields of flexible electronics [46,47,48]. We also investigated the temperature dependency of pristine and light-triggered PEDOT:PSS films. The results are presented in Figure 3H and indicate that with increasing temperature the resistance of the thin film decreases. This trend is more pronounced in the case of pristine and parallelly patterned PEDOT:PSS samples, where the interaction of PEDOT and PSS chains (and intrinsic water) is sensitive to ambient temperature and determines the electrical properties of thin films [49]. Oppositely, significantly lower temperature changes were observed in the case of perpendicularly patterned films, where the charge transfer occurred through the less sensitive to ambient temperature PEDOT-rich lines, connected to the electrodes.

Since the PEDOT:PSS represents a polymer blend that consists of the conducting PEDOT and the dielectric PSS, the charge transport in PEDOT:PSS-NAP should depend strongly on the phase separation between the two components, PEDOT and PSS. To investigate this phenomenon, we performed a series of conductive c-AFM measurements (for this purpose the PEDOT:PSS-NAP films were deposited on the conductive substrates). These measurements revealed the surface morphology and the distribution of conductive PEDOT area in the PSS matrix (Figure 4). The c-AFM maps were taken from lithographic-mask screened (Figure 4A) and illuminated sample areas (Figure 4B). As is evident, the light-triggering affects both the surface morphology and the conductivity. In particular, the illuminated areas show slightly smoothed surfaces compared to non-illuminated ones (Figure 4 top parts). Very pronounced changes were observed in the case of conductivity maps (Figure 4 bottom parts). The illuminated area exhibits the almost fully conductive surface, which corresponds to the pristine PEDOT, with only minimal PSS admixture. Oppositely, the screened area shows only several conductive spots, and in this case, an excess of PSS, transferred from illuminated areas, significantly dampens the surface conductivity. So, the results of c-AFM measurements also prove the proposed mechanism of polymers’ mutual redistribution under the light illumination, resulting in the appearance of conductive pathways and pronounced lateral electrical anisotropy. It should also be noted, that most of the previous work, aimed on the creation of anisotropic conductivity in the un-doped CPs structures and films are aimed on the utilization of phenomena related to spatial confinement of materials (in-plane and out-of-plane conductivity anisotropy) or mechanical stretching of thin films with the corresponding orientations of the polymer chains. In this work, the utilization of light-triggering for creation of anisotropic conductive behavior in thin PEDOT:PSS films was demonstrated for the first time. The results convincingly demonstrate the success of the proposed approach, but further utilization of the structures we created in areas where the electric anisotropy is desired (ranging from optoelectronics devices up to tissue engineering [50,51]) will require the introduction of the greatest anisotropic degree, which will be the aim of our further studies.

## 4. Conclusions

A new method for light-induced phase separation in PEDOT:PSS films was described and verified by different tests. The proposed approach includes the deposition of PEDOT:PSS on the flexible PLLA substrates and grafting of photoisomerisable 4-nitroazophenylene moieties to PSS chains. Subsequent light illumination of light-sensitive thin films, created in this way, at the wavelength corresponding to *E*/*Z* isomerization of 4-nitroazophenylene moieties through the lithographic mask leads to the formation of isomeric concentration gradients, which in turn induces the migration of PSS chains from the illuminated areas. As a result, a PEDOT-rich (illuminated) and PEDOT-poor (screened by mask neighbors) line array arose. The electrical conductivity of the patterned films was measured in two directions: along the created PEDOT lines and perpendicular to them. The apparent increase in conductivity was observed in the direction parallel to PEDOT lines, which provided conductive paths. In the perpendicular direction, however, the conductive paths are separated by isolating ones and the conductivity is much lower. By the present method the flexible thin films with apparent anisotropic conductivity are created, which can find applications in a range of electronic and optoelectronic devices. The proposed method, for the first time, introduces the principle of light–matter interaction for the creation of anisotropic electrical properties in the fields of conductive polymers and is compatible with the common photolithographic approach, which makes it technically attractive.

## Figures and Tables

**Figure 1 polymers-11-01856-f001:**
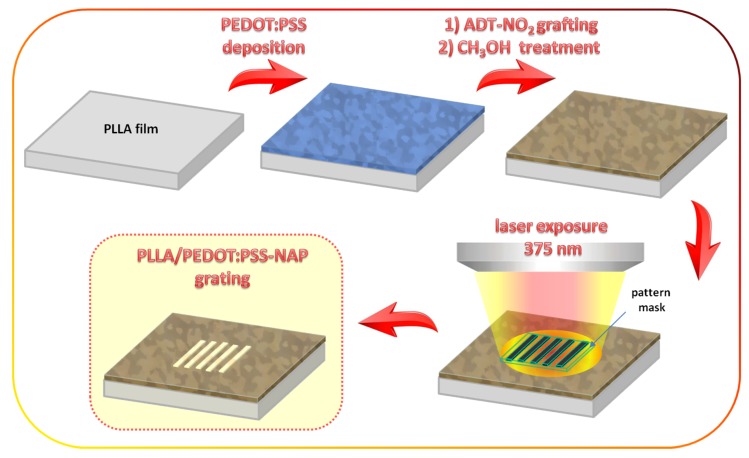
Schematic representation of experimental concept: thin PEDOT:PSS film was deposited on flexible PLLA substrate, modified by ADT-NO_2_ and treated in vapor of methanol; sample illumination with light-emitting diode (LED) was performed through the lithographic mask.

**Figure 2 polymers-11-01856-f002:**
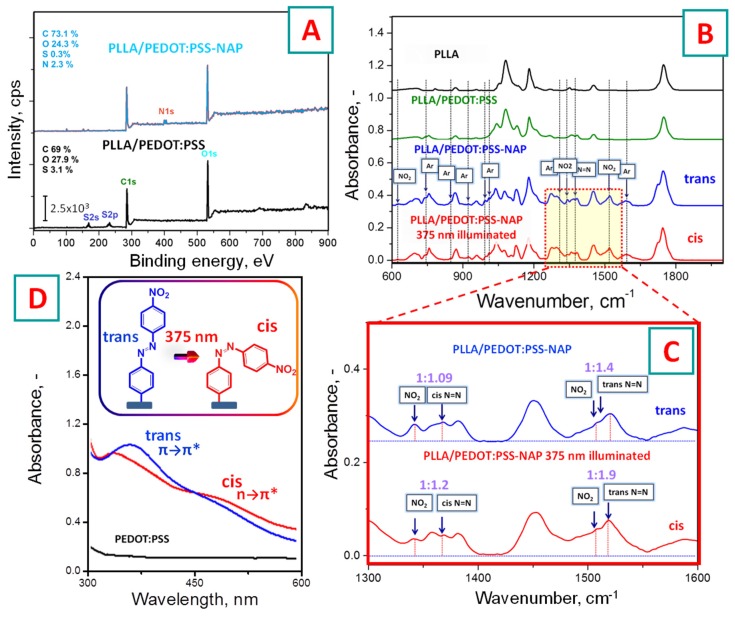
**(A**) XPS spectra of pristine PEDOT:PSS and PEDOT:PSS-NAP films deposited on PLLA substrate; (**B**) FTIR spectra measured on pristine PLLA substrate, PLLA substrate with PEDOT:PSS, and PEDOT:PSS-NAP films before and after illumination with 375 nm LED wavelength; (**C**) details of the IR spectrums of PEDOT:PSS-NAP film before and after light triggering; (**D**) illumination-induced changes in UV-Vis absorption spectra of PEDOT:PSS-NAP.

**Figure 3 polymers-11-01856-f003:**
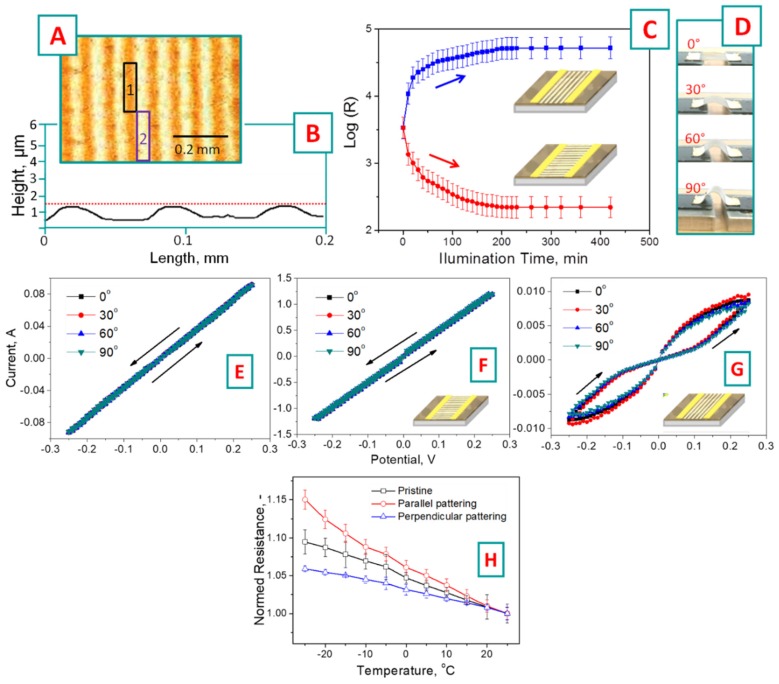
(**A**) Digital photography of PEDOT:PSS-NAP grating; (**B**) surface morphology profile, created by light-triggering through the mask (profilometry); (**C**) evolution of PEDOT:PSS-NAP, thin-film electronic resistivity during the illumination via linear mask, deposited parallel or perpendicular to electrodes; (**D**) schematic representation of bending tests. Current-voltage curves of: (**E**) pristine, (**F**) patterned perpendicularly, and (**G**) parallel patterned, thin PEDOT:PSS-NAP films under various angles of bending; (**H**) dependency of pristine and patterned samples on temperature.

**Figure 4 polymers-11-01856-f004:**
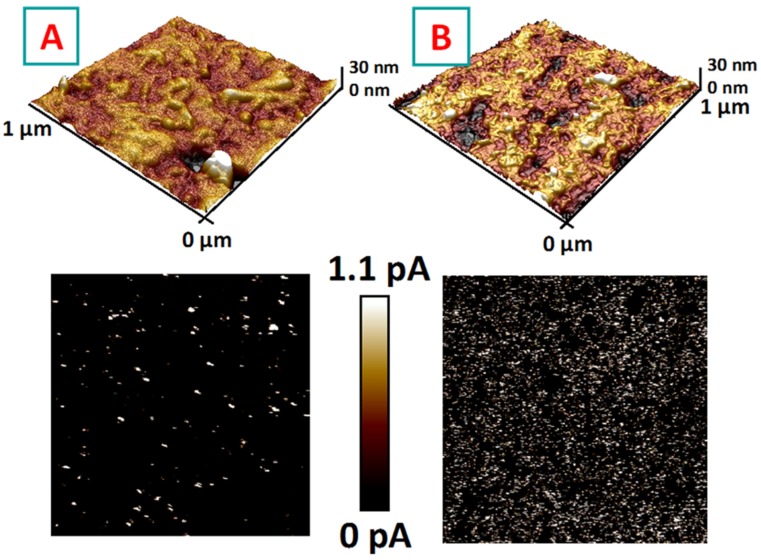
AFM morphology and conductivity scans measured on screened (**A**) and illuminated (**B**) areas of patterned PEDOT:PSS-NAP thin film.

**Table 1 polymers-11-01856-t001:** Band position of same groups in IR spectra of PEDOT:PSS-NAP samples [41].

Band Position, cm^−1^	Assignment
1746	C=O in PLLA
1587	Ar ring
1520	N=N trans
1507	NO_2_ str
1450	CH_2_ vib in PLLA
1381	SO_2_ str in PSS
1367	N=N trans
1341	CH_2_ def in PLLA
1294	NO_2_ str
1272	S=O vib in PEDOT
1178	O–C–O str in PLLA
1125	O–C–O str in PEDOT
1075	C–O str in PLLA
1038	O–S–O str in PSS
1008	Ar ring
992	Ar ring
957	C–S stretch in PEDOT
923	Ar ring
866	CH_2_ def in PLLA
756	CH_2_ def in PLLA
738	Ar ring
699	Ar ring
676	C–S stretch in PEDOT

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
