# Peer review of "Flexible Conductive Polymer Film Grafted with Azo-Moieties and Patterned by Light Illumination with Anisotropic Conductivity"

_polymers, 2019, doi:10.3390/polym11111856_

Round 1
Reviewer 1 Report
Th

Author Response
Please, see the attached file

Reviewer 2 Report
Kalachyova et al. demonstrated a method that enables anisotropic conductivity in a PEDOT:PSS thin film by light-induced polymer phase redistribution. This method is simple and straightforward. I would recommend the publication of this paper in Polymers after minor revisions. Here are some comments for the author to consider.
The height difference between peak (PSS rich area) and valley (PEDOT rich area) in Figure 3b is not clear. Please put more tick labels on the vertical axis. Discussions on the height differences should be provided. It can be seen in Figure 3g that the IV curves is quite different for samples with parallel arrangement of electrodes and light-patterns compared with that of Figure 3e and Figure 3f. There is an open area between forward scan and backward scan. Please discuss more on this. On line 195, "Figure 3H" should be "Figure 3G". Some discussions on how light induces the migration of PSS should be provided. "Bending test" were misspelled as "blending test" a few times in the manuscript, please revise. The labels in Figure 4 might be wrong. "A" should be "B" and "B" should be "A".Author Response
Please, see the attached file

Round 2
Reviewer 1 Report
See the attached file
